# Knowledge, practices and perceptions of communities during a malaria larviciding randomized trial in the city of Yaoundé, Cameroon

**Carmène Sandra Ngadjeu**[1,2ʘ], **Abdou Talipouo**[1,2ʘ], **Sévilor Kekeunou**[2],
**Patricia Doumbe-Belisse**[1,2], **Idriss Nasser Ngangue-Siewe**[1,3],
**Landre Djamouko-Djonkam**[1,4], **Edmond Kopya**[1,2], **Roland Bamou**[1,4],
**Nadège Sonhafouo-Chiana**[1,5], **Leslie Nkahe**[1,2], **Metoh Theresia Njuabe**[6],
**Parfait Awono-Ambene**[1], **Charles Sinclair Wondji**[7], **Christophe Antonio-Nkondjio**[1,7] *

1 Laboratoire de Recherche sur le Paludisme, Organisation de Coordination pour la lutte Contre les Endémies en Afrique Centrale (OCEAC), Yaoundé, Cameroun, 2 Faculty of Sciences, University of Yaoundé I, Yaoundé, Cameroon, 3 Faculty of Sciences, Department of Animal Biology, University of Douala, Douala, Cameroon, 4 Vector Borne Diseases Laboratory of the Biology and Applied Ecology Research Unit (VBID-URBEA), Department of Animal Biology, Faculty of Science of the University of Dschang, Dschang, Cameroon, 5 Faculty of Sciences, University of Buea, Buea, Cameroon, 6 Laboratory of biochemistry, University of Bamenda, Bamenda, Cameroon, 7 Vector Group Liverpool School of Tropical medicine Pembroke Place, Liverpool, United Kingdom

ʘ These authors contributed equally to this work.
* antonio_nk@yahoo.fr

**Data Availability Statement:** All relevant data are within the paper and its Supporting Information files.

# Abstract

## Background

Urban malaria is becoming a major public health concern in major cities in Cameroon. To improve malaria vector control, a pilot larviciding trial was conducted to assess its impact on mosquito density and malaria transmission intensity in Yaoundé. The present study investigated perceptions and practices of communities on malaria control during the larviciding trial implemented in Yaoundé.

## Methods

Quantitative and qualitative data were collected in non-intervention and intervention areas. Quantitative data were collected during three cross-sectional surveys using a structured pre-tested questionnaire while qualitative data were obtained through interviews. A total of 26 in-depth interviews and eight focus group discussions with community members were performed. A binary logistic regression model was used to assess the perception of the community on larviciding impact on some malaria or bed nets use indicators.

## Results

People living in intervention areas were 2.64 times more likely to know the mode of malaria transmission (95% CI: 1.82–3.84; p<0.001) and 1.3 time more likely to know mosquito

**Funding:** This work received financial support from Wellcome Trust Senior Fellowship in Public Health and Tropical Medicine (202687/Z/16/Z) to CAN. The funding body did not have any role in the design, collection of data, analysis and interpretation of data and in writing of the manuscript.

**Competing interests:** The authors have declared that no competing interests exist.

**Abbreviations:** LLINs, Long Lasting Insecticidal Nets; ITNs, Indoor Treated Nets; FGD, Focus Group Discussion.

breeding habitats (95% CI: 1.06–1.56; p = 0.009) compared to those living in non-intervention areas. In intervention areas, interviewee opinions on larviciding were generally good i.e. most interviewees reported having noticed a reduction in mosquito nuisance and malaria cases following larviciding implementation; whereas in non-intervention areas no report of reduction of mosquito nuisance was recorded. LLINs were regularly used by the population despite the implementation of larviciding treatments. There was high interest in larviciding program and demand for continuation, even if this needs the community involvement.

## Conclusion

The larviciding program in the city of Yaoundé did not negatively affected community members' behaviour and practices concerning the use of treated nets. The study indicated the acceptance of larviciding program by the population. This positive environment could favour the implementation of future antilarval control activities in the city of Yaoundé.

## Background

Malaria still remains a major public health problem in Cameroon with 3.3 to 3.7 million suspected cases recorded per year [1]. The disease affects the whole country with however large differences between regions [2]. Malaria prevention in the country relies on the use of Long-Lasting Insecticidal Nets (LLINs). Since 2004, four important free distribution campaigns of treated bed nets have been conducted in the country. LLINs ownership by the population is estimated at 73% and the usage rate at 58% [3]. Although the objective of the Ministry of Public Health (> 80% of the households having one net for two persons) is not yet achieved. It is estimated that between 2000 and 2015, the increase use of treated bed nets by Cameroonians led to a significant decrease in the prevalence of malaria reported cases from 41% to 24.3%, and 54% of malaria mortality rate [2, 4].

To achieve and maintain malaria elimination, it is crucial to sustain high coverage level of vector control measures [5]. In addition to LLINs or IRS, antilarval control measures could also be used to reduce vector densities and malaria transmission [6–8]. Larviciding through the use of microbial larvicides has proven to be largely effective for controlling malaria transmission in different epidemiological settings [7, 9, 10]. This measure is particularly indicated in places where breeding habitats are few, fix and findable such as cities [11]. Because larviciding target mosquitoes at the larval stage, it affects indoor and outdoor biting mosquitoes also mosquitoes displaying high insecticide resistance and could be a good supplement to LLINs use [6, 8]. The intervention is considered to be cost-effective with the cost per person being similar to those of LLINs in urban settings [12–14]. Larviciding compare to LLINs is not influenced by personal behaviour or frequency of usage but is deployed at the community level. However because larviciding is labour-intensive, it requires active community participation and also consent, involvement and cooperation of local communities [15–17]. As a result, gathering information on people knowledges, practices and adherence to larviciding is crucial for a successful implementation of the intervention. There have been so far not many studies assessing communities adherence and perception on larviciding trial across the continent [15, 17–19]. Studies in East Africa (Tanzania) suggested that although community members indicated their overall support to larviciding activities, they expressed neutral perceptions towards positive aspects of larviciding [20]. This attitude was mainly driven by insufficient knowledge

on malaria vectors breeding habitats, the way to carry larvicide application, the inadequate monitoring of the programme effectiveness and the lack of financial means and personal protective equipment [20]. A recent study conducted in West Africa indicated that the acceptance of larviciding in village communities was high and the majority of respondents reported a relief in mosquito nuisance and malaria episodes during larviciding implementation [19]. The perceive success of decreasing malaria episodes and mosquitoes nuisance could be a key determinant of the adhesion of communities to antilarval control measures [21, 22]. It is also not known whether significant decrease of malaria transmission could affect risks perception by local communities, as well as stakeholders support to vector control activities [23, 24].

In the present study, we tested the hypothesis that the implementation of a larviciding trial in the city of Yaoundé will improve knowledge and practices relating to malaria and mosquitoes, as well as increase people's positive perceptions towards larviciding. The study main objective was then to compare the knowledge, attitude, practices and perceptions of people living in the intervention versus non intervention areas. The study also permitted to assess additional factors such as the impact of larviciding on LLINs use; malaria/mosquito knowledge and practices; attitudes towards the intervention; and willingness to participate in larvicidal treatment. Baseline data collected in 2017 has already been published [25].

## Methods

### Study sites

The present study was undertaken in Yaoundé (3˚51'N 11˚29'E), the capital city of Cameroon. The city has about 3 million inhabitants and is the second largest city of Cameroon [26]. It is characterised by a typical equatorial climate with two rainy seasons extending from March to June and from September to November. Situated about 800 m above sea level, Yaoundé receives annually over 1700 mm of rainfall. Malaria transmission in Yaoundé is year-round, sustained by *An. gambiae*, *An. coluzzii* and *An. funestus*, reaching 92 infective bites per person per year [27, 28]. A recent study indicated bed nets ownership rate (one LLIN per household) of over 99% and usage rate of 76.7% [25]. The present study took place in 26 districts (referred to as clusters) of the city.

### Study design and larviciding activities

The primary objective of the trial was to assess the effect of larviciding on anopheline mosquito densities and malaria transmission rate in Yaoundé. A parallel cluster randomised trial was conducted in twenty-six districts referred to as clusters. Thirteen clusters served as control whereas the thirteen remaining were the intervention areas. Each cluster was an area of 2 to 4 km$^2$ crossed by a river system encompassing both lowland and highland areas. The evaluation zone was situated at the center of each cluster always in the lowland area. Clusters were separated from one another by a distance of 500 m to 1 km to minimize mosquito spillover from non-intervention to intervention areas. Administrative boundaries (including roads, railways) were used to determine the limit of clusters. Baseline entomological data were collected from all clusters for 17 months, from March 2017 to July 2018. At the end of the baseline sampling period, all clusters were ranked according to the primary outcome (adult anophelines biting densities). Clusters with similar biting densities were grouped into pairs and from each pair, one cluster was randomly selected as the intervention area and the other as control using a computer-assisted programme [25]. After baseline surveys, microbial larvicide was applied in 13 clusters for 27 months (September 2018 to November 2020) (Figure 4). Adult biting densities collected using CDC light traps were used as the primary outcome.

During the larviciding trial in the intervention sites, all aquatic habitats (with or without mosquito larvae) were identified and treated, using granules of the biological larvicide Vecto-Max G containing both *Bacillus thuringiensis* (Bti) and *B. sphaericus* as active ingredients. All aquatic habitats were treated once every two weeks by hand application at a dosage of 500 to 1500 mg/m$^2$ [29]. A team of two to three persons were involved in the treatment of breeding sites whereas a different team of two persons were involved in the evaluation of the treatment 24 to 48 hours after each treatment.

## Questionnaires, in-depth interviews and focus group discussions

The larviciding program in the city of Yaoundé was launched in 2017 with the end goal of providing critical information which could be useful in adopting larviciding as a complementary approach to control malaria transmission in major cities of Cameroon [2, 30].

**Quantitative study.**   From 2017 to 2020, three community-based descriptive cross-sectional household surveys were conducted. A pre-tested questionnaire was used to assess the population knowledges on malaria, its vector and prevention measures [25]. During the second and third survey rounds, which coincided with larviciding treatments, the questionnaire was slightly modified by adding a section on « larviciding trial » to capture people's attitudes and perceptions following the implementation of larviciding activities across the city. For each survey, a minimum of 50 households per cluster were selected for the interview. The following strategy was used for house selection, 5 houses were chosen randomly from a bloc of 30 to 50 houses for house survey. Houses enrolled were distant from one another by a minimum distance of 20m. If consent was not obtained from a selected household a neighbouring household was chosen. Different households were interviewed each year. Parents (household head, spouse or an elder representative of the house) who consented to the study were interviewed. Interviews were undertaken in French or English and in private to reduce the influence from other people. Informations collected at baseline included (i) people's knowledges and attitudes on preventive measures; (ii) ownerships and usage of LLINs; and (iii) prevention measures reported as undertaken by the population. During larviciding intervention, a section on « larviciding trial » was added to assess: (i) people's awareness of larviciding (ii) perception of larviciding impact and (iii) willingness to contribute. Comparison were conducted to assess whether the variables measured differ between the intervention and non-intervention areas at baseline and during the intervention. A total of 4,795 households participated to the study (Table 1). To assess the knowledge of respondents on malaria, the answers to four different questions including malaria signs and symptoms, mode of transmission, measures of prevention and knowledge of mosquito breeding habitats were combined. Participants providing correct answers to at least three of the questions were considered as having a correct knowledge of malaria. Those scoring less than 3 correct answers were considered as having non-correct knowledge of malaria. Concerning effective practices in regard to malaria prevention and treatment, the answers to four different questions including sleeping under a treated bed nets regularly, going to hospital for malaria treatment, eliminating standing water bodies around

**Table 1. Surveys conducted to assess population knowledge and practices.**

| Type of survey | Characteristics of study population | Number undertaken per year | | |
| --- | --- | --- | --- | --- |
| | | 2017 | 2019 | 2020 |
| Questionnaires | Number of households interviewed | 1643 | 1622 | 1530 |
| Focus group discussion | focus group/cluster (community members) | / | / | 8 focus group discussion (4 to 5 peoples per group) |
| In-depth interviews | head of districts or group leaders | / | / | 26 peoples interviewed |

houses and purchasing drugs in the pharmacy were assessed. 'people who reported engaging in less than three of these practices were considered to not be practicing effective malaria prevention whereas those engaging in three or more of these practices were considered as practicing effective malaria prevention.

**Qualitative study.** During the third survey round qualitative data were collected through in-depth interviews (IDI) with community leaders and focus group discussions (FGD) with community members. For this purpose, a semi-structured interview guide was used. The study team used a uniform discussion guide during the in-depth interviews and focus group discussion sessions which included discussion on general knowledge about malaria, vector control, and larviciding (awareness about larviciding program, perception of the impact of larviciding program and willingness to contribute to the larviciding program). A total of 26 in-depth interviews and 8 focus group discussions were conducted in both the intervention and the non-intervention areas (Table 1). For in-depth interviews, they were voice recorded with participant consent and lasted about 20 minutes, key stakeholders in the community were identified and interviewed. The study team visited every key stakeholder before the interview day to explain the purpose and objectives of the study. Key stakeholders included head of districts, religious leaders, and youth or women leaders. These people generally interact with many people in the district on a daily basis. Therefore, key stakeholders were well informed on the population perception of the larviciding program.

Focus group discussions were conducted in each district and lasted about 01 hour. All residents aged 18 years and above consenting to take part to the study were eligible to join the FGDs. Participants for the focus group discussion were selected from existing socio demographic groups present in each district(male, female, youth groups).

## Data analysis

Household surveys data were registered into Microsoft Excel database and checked for quality assurance. The data were analysed using Social Package for Social Sciences (SPSS) version 20 statistical software package. Percentages were compared using Chi squared test. The main outcomes evaluated were the evolution of the following indicators during larviciding treatments: (i) population perception about mosquito's densities, (ii) population perception about malaria cases, (iii) population adherence to larviciding treatments (iv) proportion of population that used LLIN the previous night; (v) respondent knowledge about malaria transmission, and (vi) respondents knowledge on mosquito breeding sites. To assess the impact of larviciding intervention on malaria-related knowledge and practice parameters, a binary logistic regression model was used. The classification into 'correct knowledge' or 'non correct knowledge' or 'effective practice' or 'non effective practice' was used as explanatory variable and the year and the status of the site as 'intervention' or 'non-intervention' was used as independant variables. A second binary logistic regression model was applied this time using the status of the cluster as 'intervention' and 'non-intervention' as explanatory variable and knowledge of malaria symptoms, knowledge of malaria transmission, knowledge of mosquitoes breeding sites and bed net usage as independent variable. Statistical significance was set at $P<0.05$. Data from in-depth interviews and focus group discussions were transcribed by the same researchers who conducted the interviews and key findings were summarised. The knowledge and practices of respondents on malaria control were analysed using criteria defined in the first part of the study [25]. To ensure the robustness of data analysis, the team conducting interviews hadinduction meetings and pilot interviews. An integrated interview protocol was used. Data collected were analysed at different levels and kept for future reference.

Community participation or adherence to the trial was measured as the willingness of members of the community to give access to their private properties for larviciding activities, open their homes for mosquito collection or for the storage of material used for larviciding and proposing some members to work with the team conducting larviciding treatments, adult mosquito collection or inspection of breeding sites.

## Results

### A. Quantitative data analysis

**Sociodemographic characteristics of households surveyed.** During the larviciding intervention, a total of 3,152 households were visited, including 1,622 at midway (2019) and 1,530 at the end (2020) of the study. Results of the beginning of the study were already published by Talipouo et al. [25] where 1,643 households were visited. The number of male and female taking part in the study was non significantly different ($\chi^2 = 0.028$; P = 0.87). There was about 6 people on average per household. The interviewees were aged between 17 and 55 years. The highest level of education achieved by the majority of respondents was the secondary level, followed by university and primary levels, regardless of the phase of the study and the status of the district. The majority of houses were of modern style ($\chi^2 = 831.64$; P<0.0001) and connected to the national water distribution system (Camwater) ($\chi^2 = 212.647$; P<0.0001) (Table 2).

**ITN ownership, access and usage.** Levels of LLINs ownership, access and usage at baseline and during the larviciding intervention were compared with baseline value collected in 2017 (Table 3). There was no statistical difference concerning ITN indicators measured between households located in the intervention and non-intervention areas before and during the intervention period. However, Chi-square tests revealed significant differences in bed nets

**Table 2. Characteristics of households surveyed stratified by larviciding phase and intervention status.**

| | | Baseline [a] | | Intervention | | | |
|---|---|---|---|---|---|---|---|
| Years | | 2017 | | 2019 | | 2020 | |
| Characteristics | Variables | Non intervention | Intervention | Non intervention | Intervention | Non intervention | intervention |
| **Number of households surveyed** | | **971** | **672** | **826** | **796** | **763** | **767** |
| **Gender** | Male | 33.4% | 39.1% | 44.4% | 41.4% | 46.5% | 42.6% |
| | Female | 66.6% | 60.9% | 55.6% | 58.6% | 53.5% | 57.4% |
| **Mean number of people/household** | | 5.8 | 5.4 | 5.7 | 5.7 | 5.9 | 5.8 |
| **Mean number of Children<5 years/household** | | 1.3 | 1.2 | 1.2 | 1.2 | 1.1 | 1.1 |
| **Highest education level head of household** | University | 20.9% | 25.6% | 21.4% | 28.9% | 18.6% | 23.5% |
| | Secondary | 59.8% | 56.8% | 61.6% | 55.1% | 66.0% | 58.6% |
| | Primary | 19.3% | 17.6% | 17.0% | 16.0% | 15.4% | 17.9% |
| **Occupation of the head of household** | Formal sector | 26.4% | 23.3% | 15.2% | 15.8% | 16.3% | 17.8% |
| | Informal sector | 56.3% | 56.5% | 70.4% | 60.9% | 69.0% | 61.5% |
| | Housewife | 12.7% | 9.7% | 10.8% | 14.8% | 11.7% | 14.7% |
| | Student | 4.6% | 10.5% | 3.6% | 8.4% | 3.0% | 5.9% |
| **Type of house construction** | Modern houses | 65.4% | 72.8% | 64.6% | 70.8% | 69.5% | 79.9% |
| | Traditional houses | 34.6% | 27.2% | 35.4% | 29.2% | 30.5% | 20.1% |

[a]Data published [25]

**Table 3. Ownership, access and usage of LLINs before and during the larviciding intervention.**

| Periods | | | Baseline[a] | Intervention | | | |
|---|---|---|---|---|---|---|---|
| Years | | | **2017** | **2019** | | **2020** | |
| LLINs indicators | Non intervention | Intervention | Non intervention | Intervention | Non intervention | intervention |
| % HH owning > 1 LLIN* | 99.9 | 99.7 | 97.1 | 95.7 | 93.7 | 94.0 |
| % HH owning > 1 LLIN for 2 people* | 57.5 | 59.3 | 47.8 | 44.9 | 48.9 | 50.1 |
| % population with access to LLIN* | 78.4 | 77.0 | 66.0 | 71.1 | 69.2 | 69.0 |
| % population that use LLIN previous night* | 73.1 | 73.5 | 63.8 | 68.1 | 62.4 | 64.6 |

*Significant variation between intervention and non-intervention areas compared to baseline data;

[a]Data published [25]

ownership and usage in intervention and non-intervention areas compared to baseline data. All the indicators significantly decreased with time (P<0.001) compared to the baseline data.

**Impact of larviciding intervention on malaria-related knowledge and practices.** Population knowledge and practices about malaria between non-intervention and intervention areas were compared according to periods. It appeared that people had similar knowledges and practices in both sites regardless of the period (P>0.05). However in 2019 people interviewed in intervention areas had correct knowledge and effective practices about malaria than those from the non-intervention area (P = 0.02) (Table 4).

A binary logistic regression model was used to assess any association between larviciding implementation and people knowledge and practices. From the analysis, it appeared that larviciding intervention was associated with improved knowledge of mosquito breeding habitats and mosquito role as malaria vector. 'the odds of people having correct knowledge of malaria and mosquito bionomic was 2.64 (95% CI: 1.82–3.84; p<0.001) times higher in the intervention area compared to the non-intervention area. Also, the odds of people having non-correct knowledge of mosquito breeding habitats was 1.3 time (95% CI: 1.06–1.56; p = 0.009) higher in the intervention area compared to the non-intervention area. No change in the bed net usage as a result of the larviciding intervention was recorded (Table 5).

**Table 4. Correct knowledge and practices about malaria during the implementation of the larviciding trial in Yaoundé.**

| Years | Factors | Categories | Nonintervention | Intervention | OR(95%CI) | P value |
|---|---|---|---|---|---|---|
| | | | N (%) | N (%) | | |
| 2017 [a] | Correct knowledge | No | 122 (12.56) | 73 (10.86) | 1 | |
| | | Yes | 849 (87.44) | 599 (89.14) | 0.85 (0.62–1.15) | 0.29 |
| | Effective practices | No | 939 (96.70) | 648 (96.43) | 1 | |
| | | Yes | 32 (3.30) | 24 (3.57) | 0.92 (0.54–1.58) | 0.76 |
| 2019 | Correct knowledge | No | 92 (11.12) | 61 (7.66) | 1 | |
| | | Yes | 735 (88.88) | 735 (92.34) | 0.66 (0.47–0.93) | 0.02 |
| | Effective practices | No | 631(76.30) | 567 (71.23) | 1 | |
| | | Yes | 196(27.70) | 229 (28.77) | 0.77 (0.62–0.96) | 0.02 |
| 2020 | Correct knowledge | No | 88 (11.52) | 96 (12.5) | 1 | |
| | | Yes | 676 (88.48) | 672 (87.5) | 1.1 (0.81–1.24) | 0.55 |
| | Effective practices | No | 452 (59.16) | 428 (55.73) | 1 | |
| | | Yes | 312 (40.84) | 340 (44.27) | 0.87 (0.71–1.06) | 0.17 |

N (%) = Number (Percentage); OR (95%CI) = Odds ratio (95% Confidence Interval;

[a] Data published [25]

**Table 5. Estimates of larviciding impact on key malaria-related knowledge and practice parameters.**

| Outcomes | Intervention vs Non-intervention areas | | |
| --- | --- | --- | --- |
| | Odds ratio | 95% CI | P-value |
| Knowledge of malaria symptoms | 1.01 | 0.84–1.21 | 0.87 |
| Knowledge of malaria transmission | 2.64 | 1.82–3.84 | <0.001 |
| Knowledge of mosquito breeding sites | 1.30 | 1.06–1.56 | 0.009 |
| Bed net usage | 1.30 | 0.93–1.82 | 0.12 |

## B. Qualitative data analysis

**Usage of malaria prevention and control tools.** The main malaria prevention method reported by half of participants was the use of LLINs. Other measures reported by some participants were environmental management and community sensitization. Drugs were also mentioned by few people as compound which can be used for malaria control.

*"The use of mosquito nets is the first line method to fight against malaria (Man, non-intervention cluster, IDI)"*

*"We must clean, gutters, spray insecticides, keep our close environment clean, because even if we use mosquito nets, mosquitoes are still going to bite us (Man, intervention cluster, FGD)"*

Vaccine was mentioned as the ultimate method for malaria elimination.

*"Really, we need a vaccine, I think that scientists have already found a vaccine but they refuse to give it to the population (Man, intervention cluster, IDI)".*

*"To really fight against malaria disease we need more than what you have already done, something as a vaccine could be key (Man, intervention cluster, FGD)"*

Some people mentioned the fact that it will be better to find a product that we could spray inside and outside houses as during the last century. They also talked about finding a compound that we could spray in standing water bodies to kill larvae.

*"Malaria is a resistant disease despite all the drugs found. According to me, it is better to find a product that we could spray in stagnant water to kill mosquito larvae (Man, non -intervention cluster, FGD)".*

**Opinions on the larviciding program.** In intervention districts, the majority of people interviewed were aware of the larviciding program.

*"Yes I know this program very well, the personnel carrying the treatment of breeding sites with larvicide always visit my neighbourhood. They were here yesterday (Woman, intervention cluster, FGD)"*

Few participants reported not being aware of the larviciding program.

*"I don't know the existence of the larviciding program may be because I always leave the house early in the morning for work and I come back very late. Today you saw me at home because it's my day off (Man, intervention cluster, IDI)".*

Concerning the usefulness of the program, the large majority of respondents reported that the program was a good initiative because it could help reduce mosquito densities and malaria cases.

*"It's a good intervention, they spray product in all stagnant water, so it drastically reduces mosquito densities (woman, intervention cluster, IDI)".*

*"Before, when we look at stagnant water bodies, we usually saw mosquito larvae at the surface, but now it is difficult to see them (Man, intervention cluster, FGD)".*

Some people doubt the fact that the larviciding program will reduce malaria cases due to the persistence of breeding sites and high mosquito density.

*"The treatment of breeding habitats will not be efficient because mosquito densities in our district are too high compared to other areas (Man, intervention cluster, FGD)".*

*"To decrease malaria cases, we have to destroy all stagnant water because as long as there is a standing water collection, there will be mosquitoes (Man, intervention cluster, FGD)".*

Others considered the program as a supplementary method which could complement the use of mosquito nets.

*"It works, but we have to continue using LLINs, because we have been using LLINs for so long (Man, intervention cluster, IDI)".*

According to some others, the specificity of the larviciding program was that the product does not kill other non-target organisms.

*"It decreased mosquito population and fishs also eat it without dying (Man, intervention cluster, IDI)".*

Few participants considered the treatment non-efficient.

*"No impact was found since the begining of the program, it seems like mosquito densities have increased instead (woman, intervention cluster, FGD")*.

**How to improve the program?.** To improve the program, many people in the intervention area, thought that it is better to improve the drainage of standing water by removing dirt or mud in gutters to enable water circulation; to cut grasses around houses; to remove wastes from standing water collections before carrying larviciding treatments. This will improve the efficiency of the product. For others, it was necessary and important to involve the population in the treatment of standing water collections to ensure the success of treatment activities.

*"It is better that before spraying larvicide, the team carrying the treatment of standing water collections with larvicide remove all wastes in gutters, because as long as they would carry treatment on garbage or grass, it would not be efficient (Man, intervention cluster, FGD)".*

Other people mentioned, it is better to reduce the interval between two treatment periods. According to them, it is important to treat every day or once a week. Also, increasing the treatment staff to maximise the area to be treated was proposed as well as the treatment of toilets. They also stressed the need to keep domestic animals in enclose.

*"To improve the program, it is better to regularly treat, twice or 3 times per week specifically in dry seasons when mosquito densities are high (Man, intervention cluster, FGD)".*

*"You can increase the number of treatment staff in order to make sure that the treatment is well done and surfaces such as toilets, gardening for animals and hidden breeding sites are treated (Man, intervention cluster, FGD)".*

**Perception of the larviciding impact.** *Impact on mosquito density.* Concerning the impact of larviciding treatments on mosquito densities, the majority of participants interviewed in the intervention areas indicated that mosquito densities have decreased since the implementation of larviciding treatment.

*"Mosquito densities have decreased since you have started the larviciding treatment compared to past years (Man, intervention cluster, FGD)".*

*"Your treatment has significantly decreased mosquito densities, in the evening you do not see them flying anywhere as we saw before (Woman, intervention cluster, FGD)".*

*"Mosquitoes have decreased but in dry seasons it is serious (Woman, intervention cluster, FGD)".*

*"Your product has decreased mosquito densities, but since the beginning of COVID-19 pandemic, we have not seen you for about 2 months, so mosquitoes are coming back (Man, intervention cluster, IDI)".*

In the non-intervention sites some of the participants indicated to have noticed no significant change.

*"The number of mosquitoes remained the same because we are in the lowland area, where breeding sites persist all year round (Woman, non-intervention cluster, IDI)".*

Some of the respondents in the non-intervention areas said that, mosquito densities have increased. Others mentioned the influence of season, they remarked that during the dry season mosquito densities were high.

*"Mosquito densities have increased; you cannot sit outside without insecticide spray. Yesterday when I slept mosquito came by my mosquito nets, it is smoke that drove them out, (Woman, non-intervention cluster, IDI)".*

*Impact on malaria cases.* More than half of the participants in intervention areas were of the opinion that, malaria cases have been drastically reduced. The main reason evocated was that perhaps the product used in the program kills more mosquitoes that transmit malaria than other type of mosquitoes. Other people also indicated that children are less exposed to mosquito bites because they are on holidays and they go to the bed early.

*"Malaria cases have decreased, this year I had malaria only once whereas before I did not do 3 weeks without suffering from malaria (Man, intervention cluster, IDI)".*

*"Yes, yes the number of malaria cases has decreased. Perhaps the product kills anopheles mosquitoes that give malaria (Woman, intervention cluster, IDI)".*

*"Malaria cases have decreased, as children have stopped going to school each parent is watching after their children, who do not stay late out of the house to learn (Woman, intervention cluster, FGD)".*

However, in non-intervention areas, participants mentioned that malaria cases increased or remained the same.

*"We are frequently ill, specifically children. Once they take traditional and modern drugs, they recover, but few months after they fall sick again (Man, non-intervention cluster, IDI)".*

*"The number of malaria cases increased compared to last year (Man, non-intervention cluster, IDI)".*

*Impact on LLINs use.* In both sites, a great majority of the interviewees continued to use mosquito nets as before the program. Sleeping under mosquito nets has been integrated in their daily practices; when they sleep without bed nets, they feel something is missing. The main reason provided for the high adherence to bed nets is that in lowland areas mosquitoes are presents in high densities throughout the year. According to them, prevention is better than cure.

*"It is like a custom to us; I always sleep under a mosquito net, despite the fact that the density of mosquitoes this year is less than other years. Sometimes when you sleep without bed nets, you feel as if you are outside the house; something is missing (Man, non-intervention cluster, IDI)".*

*"Do you want to kill me? I can't even try? I cannot sleep without a mosquito net; nobody here has decreased the usage of mosquito nets. In most neighbourhood houses, mosquito nets are always installed on the bed. Moreover, we are in a lowland area; mosquitoes are abundant (Man, non-intervention cluster, IDI)".*

However in intervention sites, few people said they used mosquito less than the other years. They used nets according to the season and to mosquito densities.

*"We decreased the usage of mosquito nets. When we observe few mosquitoes, we do not use bed nets (Man, intervention cluster, IDI)".*

*Impact on the amount spent for vector control and malaria treatment.* In general, the amount spent by the majority of interviewees for vector control and malaria treatment was lowest after implementing the larviciding program in intervention sites. According to about 25% of participants the amount spent depends on the season.

*"Actually, the amount spent to fight against malaria is reduced. Before we spent two hundred (200) CFA F per day to buy repellents that we burn. Now we spent two hundred (200) CFA F every four days. (Man, intervention cluster, IDI)".*

*"I sometimes use a fan or insecticide sprays. The amount spent to fight against mosquitoes burden and malaria is reduced (Woman, intervention cluster, IDI)".*

*"Yes, we have reduced the amount spent to fight against malaria per season. In the dry season we spend more to fight against malaria than in the rainy season (man, intervention cluster, FGD,)".*

Participants in the non-intervention areas indicated that there was no change in the amount spent to fight against malaria. They spend like before the larviciding program. Larviciding treatment is considered not to have reduced their expenses.

*"Expenses remained the same. I always buy about 2 or 3 mosquito nets per year (Man, non-intervention cluster, FGD)".*

**Willingness to participate to the larviciding program.** In both sites, more than 80% of the participants agreed to freely participate in the program. In intervention sites, people mentioned knowing how to treat since they regularly assist the team conducting larviciding treatment in their neighbourhood. While in non-intervention sites, they mentioned knowing very well where to find breeding sites in their districts.

*"I will be glad to do it, because I know very well my district. I will apply the larvicide everywhere (Man, non-intervention cluster, IDI)".*

*"We are waiting for this opportunity since long now. Yes, we hope to have your support (Man, non-intervention cluster, FGD)".*

Other people in intervention areas willing to participate said they needed additional training to better know the larvicide characteristics, how it is used and right application doses for breeding sites.

*"We agree to participate, but we need some training on the storage, application and management of larvicide. (Man, intervention cluster, FGD)".*

*"I am capable of doing it. I spent most of my time in the lowland area. I have seen you doing it but you have to give us instructions on how to use the larvicide (Man, intervention cluster, IDI)".*

In both sites just few people thought they are not able to carry larviciding treatments, because they don't know how to store the product. They have children who can take it to play or eat it. Some said they do not have the necessary equipment to perform the work but if this is provided, with the support of the intervention team, they can do the job.

*"I cannot carry larviciding treatment or store the larvicide in my home because I have children in my house who can take the product to play or to eat. It is better you apply the larvicide (Man, intervention cluster, FGD)".*

*"I am not able to do it due to my work which takes almost all my time (Man, non-intervention, FGD)".*

## Discussion

The study's objective was to assess the knowledge and perceived impact of larviciding by the population living in the city of Yaoundé particularly in the intervention areas. The results indicated positive perception of the larval control intervention by the population. The high perceived level of effectiveness of microbial larvicide treatments by the population is due to the high impact of previous campaigns reported to reduce mosquito abundance and nuisance [31, 32]. During colonial period and well after, mosquito control was conducted by hygiene and

sanitation services in the two main cities of Cameroon, Douala and Yaoundé [31]. These actions based on regular inspection and destruction of all temporary larval habitats near houses and the clearing of bushes permitted significant reduction of malaria transmission risk [31, 32]. Although, the practice of antilarval control measures have received since then little attention [33], the success of these previous interventions are still fresh in the minds of people both elders and young ones. According to some interviewees, the fact that the microbial used is safe for the environment and non-target organisms guarantee the support to the intervention. Studies conducted elsewhere indicated that microbial larvicides are environmentally safe, specific in their action and highly efficient against *Anopheles* species [10, 22, 34, 35].

Most of the participants interviewed in the intervention areas acknowledged the fact that a significant reduction of mosquito biting densities and malaria cases were recorded. However few still complained that they didn't noticed any change. This result could derive from the fact that while a significant drop of *Anopheles* mosquito densities were recorded, no similar decrease was recorded for *Culex* and *Aedes* species [29]. These species were not specifically targeted in the course of the study. For instance pit latrine in private homes were not always treated by the team performing the treatments because they could not have access to them. Some authors reported increase in *Culex quinquefasciatus* population (within 1 or 2 weeks) following larviciding treatments, due to the creation of new breeding sites and probably absence of competition with anopheline species [33, 36]. Moreover, controlling other mosquito species such as *Culex* could depend on the context (sites), accessibility of breeding sites and the relative abundance of each species [22, 37]. It was reported that frequent retreatment of breeding habitats at 09 days intervals for several months could significantly reduce culicine mosquitoes biting densities [38]. Although the reduction of non-malaria vector mosquitoes may not have had any impact on malaria transmission, it however influences local communities' perception of success [19]. For most people, reducing malaria transmission risk should be associated to low mosquitoes nuisance irrespective of the species [19].

The majority of participants from the study reported being aware of the larviciding program while talking to the larviciding team. Those unaware of the intervention owed this to theirs professional activities which makes them leave their houses very early and come back late. This observation is in accordance with previous findings revealing that, direct interaction with local communities through activities on the field are more powerful communication means compared to sensitization only [19]. Also, making the treatment team dress differently from other people by putting on a combination during their duties and communicating with the community could improve communities adherence and participation to the programme.

Regarding knowledge on mosquitoes bionomic, people living in intervention areas were 2.64 times more likely to know the mode of malaria transmission and 1.3 time more aware of mosquitoes breeding habitats compared to those living in non-interventions areas. Yet no increase of "correct knowledge" or "effective practices" of inhabitants living in intervention compare to those of the non-intervention sites was recorded and clearly point to the low sensitivity of some indicators combined to assess "correct knowledge" or "effective practices" in the population. Beside the heterogeneous structure of the population interviewed (education level, economic or social status) which was not considered during analysis and might have also reduce the power of analysis. Malaria transmission in the city of Yaoundé is highly heterogeneous with districts registering high malaria transmission and others registering very low or no malaria transmission [27] this could have affected people knowledge on malaria vectors bionomic and disease transmission. Nevertheless, the present findings suggest that people leaving in intervention areas were deeply influenced by activities undertaken in their districts. The interest of local inhabitant to the implementation of antilarval control measures could be used to improve community participation in larviciding interventions. The team of larviciding

applicators was asked to always explain the purpose of the study to everyone they met, in order to facilitate access to mosquito breeding sites; many being located in private properties. As reported elsewhere [18], the continuous presence of the larviciding teams in intervention districts (twice a month) may have acted as a regular reminder of the role of mosquitoes in malaria transmission to most people living in the intervention areas.

Concerning ways to improve the program, most of the people interviewed, suggested to involve the community at different levels to ensure the success of the intervention. According to WHO, community sensitisation and participation are a pillar component of integrated vector management strategies [39]. This strategy has been reported to be effective in some areas where the National Malaria Control Program closely works with leaders of communities or districts to ensure the population support to the intervention [18]. Almost all respondents stated they are willing to contribute to future larviciding programs. Studies in East Africa also suggested high commitment of community members to undertake larviciding treatments or to contribute financially to the implementation of larviciding programs [19, 34, 40].

Some of the people interviewed indicated the need to reduce the time between two treatments to improve larviciding treatments particularly during the rainy season. Most studies conducted indicated larviciding to be more effective in the dry season compared to the rainy season [18].

The main control measure in Cameroon is LLINs. The ownership and usage of this tool in Yaoundé [25, 41] was not affected by the implementation of larviciding programme. Yet a decrease in the ownership rate and utilization was recorded in both the intervention and non-intervention areas and could be associated to damages on the nets rather than to the implementation of larviciding activities. Indeed, the last mass distribution campaign of LLINs in Yaoundé took place in 2015, some ITN could have been destroyed, or given to relatives not living in the town. Although people noticed a decrease in mosquito densities following larviciding activities they did not find it necessary to reduce the use of LLINs at night. This particular behaviour could be link to the frequent sensitisation campaigns made by the Cameroon government through different communication channels (TV, Radio, social media. . .) which may have contributed to increase adherence to this tool. It was hypothesised that if mosquito nuisance or infections decreased following larval control activities, people could no more perceive malaria as a major health threat and could therefore, abandon the use of preventive measures such as ITN; but the study turn to reject this hypothesis.

The study also presented some limits. (i) The mode of selection of clusters based on entomological indicators not considering social, economic, spatial or environmental variables may have limited the power of assessment of indicators associated with people knowledge and practices. (ii) The ranking of people knowledge or practices into correct or no correct could be interpreted in many ways. Many nuances might account for non correct or 'poor' practices. For instance people may not seek hospital treatment because they cannot afford it (or the transportation to get to it), or have had negative experiences in clinics before. This falls within limits of quantitative surveys using ranking measurements. (iii) Although some demographic parameters were collected from people taking part to the survey not all were always used in analysis, such as the education level, gender or the socio-economic status this could have mask important differences in population knowledge, and practices between the intervention and the non intervention areas.

## Conclusion

The larviciding program in the city of Yaoundé was positively perceived by the community although the impact of the programme was diversely appreciated. The study indicated high

acceptance of larviciding trial by the population and willingness to be involved in future larviciding activities. This positive environment could facilitate the implementation of future larviciding activities in the city of Yaoundé and could contribute to the success of this intervention.

## Supporting information

**S1 Data.**
(XLSX)

**S1 File. Interview guide for focus groups and in-depth interviews.**
(DOCX)

**S2 File.**
(DOCX)

**S3 File.**
(DOCX)

## Acknowledgments

### Ethics approval and consent to participate

The study was conducted under the ethical clearance N˚ 2016/11/832/CE/CNERSH/SP delivered by the Cameroon National Ethics Committee on Human health. Further consent was obtained from each head of district. Verbal and formal informed consent were obtained from all respondents and the study purpose was explained to them.

## Author Contributions

**Conceptualization:** Carmène Sandra Ngadjeu, Christophe Antonio-Nkondjio.

**Data curation:** Carmène Sandra Ngadjeu, Abdou Talipouo, Christophe Antonio-Nkondjio.

**Formal analysis:** Carmène Sandra Ngadjeu, Abdou Talipouo, Christophe Antonio-Nkondjio.

**Funding acquisition:** Christophe Antonio-Nkondjio.

**Methodology:** Carmène Sandra Ngadjeu, Abdou Talipouo, Christophe Antonio-Nkondjio.

**Supervision:** Sévilor Kekeunou, Parfait Awono-Ambene, Charles Sinclair Wondji, Christophe Antonio-Nkondjio.

**Validation:** Parfait Awono-Ambene, Charles Sinclair Wondji, Christophe Antonio-Nkondjio.

**Writing – original draft:** Carmène Sandra Ngadjeu, Abdou Talipouo.

**Writing – review & editing:** Carmène Sandra Ngadjeu, Abdou Talipouo, Sévilor Kekeunou, Patricia Doumbe-Belisse, Idriss Nasser Ngangue-Siewe, Landre Djamouko-Djonkam, Edmond Kopya, Roland Bamou, Nadège Sonhafouo-Chiana, Leslie Nkahe, Metoh Theresia Njuabe, Parfait Awono-Ambene, Charles Sinclair Wondji, Christophe Antonio-Nkondjio.

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
