## [Decision Letter · Decision Letter 0]

1 Apr 2022

PONE-D-21-27631

Knowledge, practices and perceptions of communities during a malaria larviciding randomized trial in the city of Yaoundé, Cameroon

PLOS ONE

Dear Dr. Antonio-Nkondjio,

Thank you for submitting your manuscript to PLOS ONE. After careful consideration, we feel that it has merit but does not fully meet PLOS ONE’s publication criteria as it currently stands. Therefore, we invite you to submit a revised version of the manuscript that addresses the points raised during the review process.

Please take the reviewer's comments in to consideration when revising this manuscript.

We look forward to receiving your revised manuscript.

Kind regards,

Eric Fèvre

Academic Editor

PLOS ONE

Journal Requirements:

Reviewers' comments:

Reviewer's Responses to Questions

**Comments to the Author**

1. Is the manuscript technically sound, and do the data support the conclusions?

Reviewer #1: Partly

2. Has the statistical analysis been performed appropriately and rigorously? 

Reviewer #1: No

3. Have the authors made all data underlying the findings in their manuscript fully available?

Reviewer #1: No

4. Is the manuscript presented in an intelligible fashion and written in standard English?

Reviewer #1: No

5. Review Comments to the Author

Reviewer #1: This paper describes a study on knowledge and practices relating to malaria control, in the light of an intervention – larvicidal treatment – in Yaoundé, Cameroon. The paper includes valuable data and important insights, which would add usefully to the scientific literature. However, at the moment insufficient detail is provided regarding the methodology and results to be confident in the validity of the findings.

The relationship between this manuscript and other related studies is not clear. In particular, the data included from 2017 appears to have been already published by Talipouo et al. 2019, but this is not made sufficiently clear in the manuscript.

The aims and questions/hypotheses to be tested in this paper need to be clearer. The objectives are described as to ‘evaluate people's knowledge and perception of the success of a pilot larviciding trial’. However, the results presented address several questions, including: the impact of the larvicidal intervention on LLIN use; the impact of the intervention on malaria/mosquito knowledge and practices; people’s perceptions of the intervention on mosquitoes and malaria cases; attitudes towards the intervention; and willingness to participate in larvicidal treatment. These are mostly presented as a comparison between intervention and non-intervention areas. If the aim of the paper was to assess how these metrics differ between the intervention and non-intervention areas, this should be stated clearly in the objectives and made clearer in the methodology. In the methodology make sure that it is clear which part of the study is being addressed by each part of the methods.

More detail is needed in the methodology and results in order to tell if findings are robust and meaningful. For example:

- A randomised control trial is mentioned in the title, but the processes of randomisation or assigning intervention and non-intervention areas are not explained. It appears that this control trial is described in Antonio-Nkondjio et al. 2021 (ref 30) but some detail is required here because it has important implications regarding whether the findings are robust, since it is plausible the findings could be explained by the method of selecting intervention and control areas.

- How were households selected for questionnaire administration in each cluster? Was this randomised? Were the same households visited in subsequent years? If so, this needs to be accounted for in the statistics. If not, this should be stated.

- Binary measures e.g. ‘good practice’ and ‘good knowledge’ – how were these defined, how were the questionnaire data turned into a binary outcome?

- How was the binary model constructed – what were the outcomes and explanatory variables? Was each assessed in a separate model?

- How were the qualitative data analysed? What measures were taken to ensure the robustness of the analyses?

- In the results, make sure it is clear what statistics have been used to generate the p-values indicated (for example line 220, line 231), and include test statistics. Check the wording for odds ratios – it should read ‘the odds of people having good knowledge was 2.64 times higher’ rather than people were 2.64 times more likely to be aware.

Supplementary material is mentioned but was not provided to the reviewer. The raw data and copies of the questionnaire and interview guides should be provided.

In addition the English writing needs improvement in places since there are quite a lot of minor grammatical errors.

6. PLOS authors have the option to publish the peer review history of their article (what does this mean?). If published, this will include your full peer review and any attached files.

Reviewer #1: No

---

## [Author Response · Author response to Decision Letter 0]

2 Jun 2022

To

The Editor in chief

Plos One

Sir/Madam,

I will like to thank the reviewers and editor for their comments which substantially improved the quality of the manuscript. All requested changes were undertaken accordingly.

Reviewer #1: 

Specific comments

Comment 1: The relationship between this manuscript and other related studies is not clear. In particular, the data included from 2017 appears to have been already published by Talipouo et al. 2019, but this is not made sufficiently clear in the manuscript.

Answer 1: Information included see end of the background section and information added in the legend of tables

Comment 2: The aims and questions/hypotheses to be tested in this paper need to be clearer. The objectives are described as to ‘evaluate people's knowledge and perception of the success of a pilot larviciding trial’. However, the results presented address several questions, including: the impact of the larvicidal intervention on LLIN use; the impact of the intervention on malaria/mosquito knowledge and practices; people’s perceptions of the intervention on mosquitoes and malaria cases; attitudes towards the intervention; and willingness to participate in larvicidal treatment. These are mostly presented as a comparison between intervention and non-intervention areas. If the aim of the paper was to assess how these metrics differ between the intervention and non-intervention areas, this should be stated clearly in the objectives and made clearer in the methodology. In the methodology make sure that it is clear which part of the study is being addressed by each part of the methods.

Answer 2: Corrected see in line 107-114. Further information was provided in the methodology see lines 197-215

Comment 3: A randomised control trial is mentioned in the title, but the processes of randomisation or assigning intervention and non-intervention areas are not explained. It appears that this control trial is described in Antonio-Nkondjio et al. 2021 (ref 30) but some detail is required here because it has important implications regarding whether the findings are robust, since it is plausible the findings could be explained by the method of selecting intervention and control areas.

Answer 3: We added in new section entitled selection of the intervention areas providing details on the way clusters were chosen and the study design, see lines 127-132

Comment 4: How were households selected for questionnaire administration in each cluster? Was this randomised? Were the same households visited in subsequent years? If so, this needs to be accounted for in the statistics. If not, this should be stated.

Answer 4: Corrected see in lines 143-146, the following information were added: “For each survey, a minimum of 50 households per cluster were randomly selected for the interview. Different households were interviewed each year. Parents (household head, spouse or an elder representative of the house) who consented to the study were interviewed”.

Comment 5: Binary measures e.g. ‘good practice’ and ‘good knowledge’ – how were these defined, how were the questionnaire data turned into a binary outcome?

Answer 5: Corrected see in line 154-165, the following information were added: “To assess the knowledge of respondents on malaria, the answers to four different questions including malaria signs and symptoms, mode of transmission, measures of prevention and knowledge of mosquito breeding habitats were combined. Participants providing correct answers to at least three of the questions were considered as having a good knowledge of malaria. Those who had less than 3 correct answers were considered as having poor knowledge of malaria. Concerning good practices in regard to malaria prevention and treatment, the answers to four different questions including sleeping under a treated bed nets regularly, going to hospital for malaria treatment, eliminating standing water bodies around houses and purchasing drugs in the pharmacy were assessed. Participants providing appropriate answers to at least three of the questions were considered as applying good practices while those with less than 3 corrects answers were considered having poor practices”.

Comment 6: How was the binary model constructed – what were the outcomes and explanatory variables? Was each assessed in a separate model?

Answer 6: Corrected see data analyses section, lines 206-209

Comment 7: How were the qualitative data analysed? What measures were taken to ensure the robustness of the analyses?

Answer 7: Information added see data analyses section lines 210-215

Comment 8: In the results, make sure it is clear what statistics have been used to generate the p-values indicated (for example line 220, line 231), and include test statistics

Answer 8: Corrected see results section

Comment 9: Check the wording for odds ratios – it should read ‘the odds of people having good knowledge was 2.64 times higher’ rather than people were 2.64 times more likely to be aware.

Answer 9: Changed throughout the text see lines 276-280

All requested changes were done accordingly and are highlighted in the main text. We look forward to hearing from you in due course.

Sincerely yours

---

## [Decision Letter · Decision Letter 1]

23 Aug 2022

PONE-D-21-27631R1Knowledge, practices and perceptions of communities during a malaria larviciding randomized trial in the city of Yaoundé, CameroonPLOS ONE

Dear Dr. Antonio-Nkondjio,

Thank you for submitting your manuscript to PLOS ONE. After careful consideration, we feel that it has merit but does not fully meet PLOS ONE’s publication criteria as it currently stands. Therefore, we invite you to submit a revised version of the manuscript that addresses the points raised during the review process. Please revise the paper according to the reviewer's comments, especially on improving the clarity in the data and statistical analysis. 

We look forward to receiving your revised manuscript.

Kind regards,

Guangyu Tong

Academic Editor

PLOS ONE

Additional Editor Comments (if provided):

We invited field experts to review your paper. Please address the reviewer comments accordingly, especially the clarity in the data and analysis presented in this paper.

Reviewers' comments:

Reviewer's Responses to Questions

**Comments to the Author**

1. If the authors have adequately addressed your comments raised in a previous round of review and you feel that this manuscript is now acceptable for publication, you may indicate that here to bypass the “Comments to the Author” section, enter your conflict of interest statement in the “Confidential to Editor” section, and submit your "Accept" recommendation.

Reviewer #2: (No Response)

2. Is the manuscript technically sound, and do the data support the conclusions?

Reviewer #2: Partly

3. Has the statistical analysis been performed appropriately and rigorously? 

Reviewer #2: I Don't Know

4. Have the authors made all data underlying the findings in their manuscript fully available?

Reviewer #2: Yes

5. Is the manuscript presented in an intelligible fashion and written in standard English?

Reviewer #2: Yes

6. Review Comments to the Author

Reviewer #2: 1. Addressing the hypothesis. Early, a hypothesis states a larviciding trial will improve intervention-area residing people’s knowledge and practices related to malaria and its vectors, and generate positive perceptions among them towards larviciding, when compared with people in non-intervention areas. Additionally, the study assessed whether larviciding impacted LLIN use.

In the discussion, it is concluded that people in intervention areas were more likely to know the mode of transmission and be aware of breeding habitats compared to people in non-intervention sites, and thus the aspect of ‘knowledge’ is suggested to have improved. However, looking at Table 4, it seems that the percentage of those with ‘good knowledge’ actually reduces over time in the intervention category (89.4% in 2017, 88.88% in 2019, and 87.5% in 2020), and is non linear in the non-intervention category (87.44% in 2017, 92.34% in 2019, and 88.48% in 2020). For full disclosure, I am not a quantitative researcher, and thus I may simply be missing the logic – but it is not clear to me how this translates into greater odds of people having ‘good knowledge’ following intervention than in areas where there was no intervention.

Furthermore, while the ‘knowledge’ portion of the hypothesis is addressed in the discussion, ‘good practices’ as described in the methods section (lines 159-163) to be assessed are not addressed apart from the component of bed-net usage (e.g. going to hospital, eliminating standing water, purchasing drugs from pharmacy). Looking at Table 4, it would seem that the percent of people engaging in self-reported ‘good practices’ as defined by the authors’ criteria (at least 3 practices) declined to zero in both non- and intervention areas. I think it's safe to assume this decline was not related to the intervention itself – in fact, given the very low baseline numbers, it is likely that other factors play a role (see next comment) – a line or two on this would add value in the Discussion (as has been done with the explanation for lowered bed net usage).

2. Terms: ‘poor’/‘bad’ practices. It feels crude to categorise people’s practices as either being ‘good’ or ‘poor’ (or ‘bad’) due to the many nuances which might account for ‘poor’ practices. People may not seek hospital treatment because they cannot afford it (or the transportation to get to it), or have had negative experiences in clinics before. Labelling them ‘poor’ implies ignorance or failure. Could authors state it as a limitation that the quantitative survey can’t account for the ‘why’, and that it may obscure these nuanced realities? Related to this, line 164 says participants with less than 3 ‘correct’ answers were considered to have poor practices. ‘Correct’ again implies wrongness. Can you describe this differently. For example: ‘people who reported engaging in less than three of these practices were considered to not be practicing effective malaria prevention, whether due to knowledge, capacity or other limitations’.

3. Describe intervention. Can you please describe the intervention in the Methods section? What did it entail, how did it work? Although seemingly described in previous publication, the context is important for this paper. Such a brief description could be included under the ‘Selection of intervention areas’ heading.

4. What are clusters? Under the new ‘selection of intervention areas’ section, the unit ‘clusters’ is used to describe where entomological data was collected. Can the authors define a ‘cluster’ (I assume it is a geographical unit)? Were ‘cluster pairs’ created based on any variable other than ‘biting densities’? This is important as differences in outcomes between paired intervention/non-intervention clusters could result from other variables like differences in socio-economic status or education level between clusters. If not, it should be listed as a limitation that other social, economic, and perhaps spatial/environmental variables were not considered when selecting for pairs of intervention/non-intervention clusters. I realise this comes from a previously published paper, but context is needed for readers to understand this one. From a quick look at Table 2, it seems that intervention areas might have been slightly wealthier? For instance, 79.9% of houses in intervention areas were ‘modern’ while 69.5% were modern in non-intervention areas. Similarly, a higher proportion of the population was university educated in intervention areas.

5. Community participation. It is not entirely clear what is meant by participation, or as is commonly used in the paper ‘adherence’ of community members to larviciding. If larviciding is done by external teams, how might communities ‘adhere’? Elsewhere, participants are said to ‘freely participate’. Can the authors describe how community members were expected to ‘participate’ or ‘adhere’ in the trial? This could perhaps be done in a description of what the trial actually entailed and how it worked (see above comment), with modifications made elsewhere as necessary for clarity.

6. Selection of participants. Can the authors describe more about the random selection of households for the survey? Was a population/address register used? Similarly, the paper does not state how participants for the focus groups selected although it is stated that they were selected to represent ‘different demographic groups’ (line 186). Can the authors say what demographic groups these were?

7. Qualitative data analysis. Collection of qualitative data is described, but analysis is not well described beyond transcription, and key findings summarised. Was thematic coding or analysis conducted? What is meat by ‘analysed at different levels’?

8. Demographic differences. Although some demographic data is given about participants in the survey, the analysis does not differentiate between different cohorts of people. Were there differences between women and men, those with higher levels of education and lower, or by house type? If such analyses were not performed, or outcomes not deemed significant enough to report, the authors should state as such. Or perhaps, indicate that future publications might explore these differences. Indeed, generic conclusions can mask over important differences in knowledge, practices (and capacity) and perceptions that may have significant implications for who is most affected by malaria transmission.

9. Quick clarifications. Line 86 – Sentence is not clear. What is meant by ‘the cost person’ as being ‘protected’? Perhaps the authors mean to say: The intervention is considered to be cost-effective, with costs per person being similar to those of LLIN use in urban settings’.

10. Long/confusing sentence. Line 107 – The sentence is long and confusing. I suggest rewording slightly (see below), and then removing the latter part of the sentence as the point about comparison between the intervention area and the non-intervention area is made in the next sentence. Note that ‘the larviciding trial’ should be replaced with ‘a larviciding trial’ as the specific trial being investigated in this paper has not yet been introduced or described. Suggested wording: In the present study, we tested the hypothesis that the implementation of a larviciding trial in the city of Yaoundé will improve knowledge and practices relating to malaria and mosquitoes, as well as increase people’s positive perceptions towards larviciding.

11. Quick clarification. Line 149 – the authors describe the focus of the qualitative survey. It is unclear how 1) people’s knowledge and attitudes on preventive measures is different from 3) prevention measures. Do the authors mean the preventative practices reported as being undertaken by interviewees?

7. PLOS authors have the option to publish the peer review history of their article (what does this mean?). If published, this will include your full peer review and any attached files.

Reviewer #2: No

---

## [Author Response · Author response to Decision Letter 1]

6 Sep 2022

To

The Editor in chief

Plos One

Sir/Madam,

I will like to thank the reviewer and editor for their comments which substantially improved the quality of the manuscript. All requested changes were undertaken accordingly.

Reviewer #2: 

Specific comments

Comment 1: Addressing the hypothesis. Early, a hypothesis states a larviciding trial will improve intervention-area residing people’s knowledge and practices related to malaria and its vectors, and generate positive perceptions among them towards larviciding, when compared with people in non-intervention areas. Additionally, the study assessed whether larviciding impacted LLINuse.

In the discussion, it is concluded that people in intervention areas were more likely to know the mode of transmission and be aware of breeding habitats compared to people in non-intervention sites, and thus the aspect of ‘knowledge’ is suggested to have improved. However, looking at Table 4, it seems that the percentage of those with ‘good knowledge’ actually reduces over time in the intervention category (89.4% in 2017, 88.88% in 2019, and 87.5% in 2020), and is non linear in the non-intervention category (87.44% in 2017, 92.34% in 2019, and 88.48% in 2020). For full disclosure, I am not a quantitative researcher, and thus I may simply be missing the logic – but it is not clear to me how this translates into greater odds of people having ‘good knowledge’ following intervention than in areas where there was no intervention.

Furthermore, while the ‘knowledge’ portion of the hypothesis is addressed in the discussion, ‘good practices’ as described in the methods section (lines 159-163) to be assessed are not addressed apart from the component of bed-net usage (e.g. going to hospital, eliminating standing water, purchasing drugs from pharmacy). Looking at Table 4, it would seem that the percent of people engaging in self-reported ‘good practices’ as defined by the authors’ criteria (at least 3 practices) declined to zero in both non- and intervention areas. I think it's safe to assume this decline was not related to the intervention itself – in fact, given the very low baseline numbers, it is likely that other factors play a role (see next comment) – a line or two on this would add value in the Discussion (as has been done with the explanation for lowered bed net usage).

Answer1: We will like to thank the reviewer for highlighting this part. Data in table 4 give general figures on correct knowledge and practices about malaria and a clear trend was not capture using the four different answers put together but when the analysis was splitted and analysis done for individual parameters significant differences between intervention and non-intervention sites was recorded for exemple knowledge on mode of malaria transmission and breeding habitats. The following show the difference in the sensitivity of indicators used. That’s why in table 5 we display indicators which were more sensitive and showing difference in knowledge between the intervention and non-intervention site. A paragraph discussing this have been added in the discussion. Concerning the second part of his comments on good practices some data were missen so we added this data in table 4. 

Comment 2: Terms: ‘poor’/‘bad’ practices. It feels crude to categorise people’s practices as either being ‘good’ or‘poor’ (or ‘bad’) due to the many nuances which might account for ‘poor’ practices. People may not seek hospital treatment because they cannot afford it (or the transportation to get to it), or have had negative experiences in clinics before. Labelling them ‘poor’ implies ignorance or failure. Could authors state it as a limitation that the quantitative survey can’t account for the ‘why’, and that it may obscure these nuanced realities? Related to this, line 164 says participants with less than 3 ‘correct’ answers were considered to have poor practices. ‘Correct’ again implies wrongness. Can you describe this differently. For example: ‘people who reported engaging in less than three of these practices were considered to not be practicing effective malaria prevention, whether due to knowledge, capacity or other limitations’.

Answer 2: The terms good and poor practices and knowledge were changed as requested by the reviewer in the section “quantitative study” in the material and method section. We also added the idea he mentioned in the limitation of quantitative survey in the discussion. 

Comment 3: Describe intervention. Can you please describe the intervention in the Methods section? What did it entail, how did it work? Although seemingly described in previous publication, the context is important for this paper. Such a brief description could be included under the ‘Selection of intervention areas’ heading. 

Answer 3: We added a description of the intervention. In the Method section. 

Comment 4; What are clusters? Under the new ‘selection of intervention areas’ section, the unit ‘clusters’ is used to describe where entomological data was collected. Can the authors define a ‘cluster’ (I assume is a geographical unit)? Were ‘cluster pairs’ created based on any variable other than ‘biting densities’? This is important as differences in outcomes between paired intervention/non-intervention clusters could result from other variables like differences in socio-economic status or education level between clusters. If not, it should be listed as a limitation that other social, economic, and perhaps spatial/environmental variables were not considered when selecting for pairs of intervention/non-intervention clusters. I realise this comes from a previously published paper, but context is needed for readers to understand this one. From a quick look at Table 2, it seems that intervention areas might have been slightly wealthier? For instance, 79.9% of houses in intervention areas were ‘modern’ while 69.5% were modern in non-intervention areas. Similarly, a higher proportion of the population was university educated in intervention areas.

Answer 4: We thank the reviewer for this comment yes clusters were selected according to entomological indicators (adult anophelines biting densities). Information on the study design could be found in the method section. Limits of the study were added in the discussion. 

Comment 5: Community participation. It is not entirely clear what is meant by participation, or as is commonly used in the paper ‘adherence’ of community members to larviciding. If larviciding is done by external teams, how might communities ‘adhere’? Elsewhere, participants are said to ‘freely participate’. Can the authors describe how community members were expected to ‘participate’ or ‘adhere’ in the trial? This could perhaps be done in a description of what the trial actually entailed and how it worked (see above comment), with modifications made elsewhere as necessary for clarity.

Answer 5: The following information was added in the data analysis section Community participation or adherence to the trial was measured as the willingness of members of the community to give access to their private properties for larviciding activities, open their homes for mosquito collection or for the storage of material used for larviciding and proposing some members to work with the team conducting larviciding treatments, adult mosquito collection or inspection of breeding sites.

Comment 6: Selection of participants. Can the authors describe more about the random selection of households for the survey? Was a population/address register used? Similarly, the paper does not state how participants for the focus groups selected although it is stated that they were selected to represent ‘different demographic groups’ (line 186). Can the authors say what demographic groups these were?

Answer 6: The following information were added in the section “Quantitative study”: “The following strategy was used for house selection, 5 houses were chosen randomly from a bloc of 30 to 50 houses for house survey. Houses enrolled were distant from one another by a minimum distance of 25m. If consent was not obtained from a selected household a neighbouring household was chosen. “. 

The following information was added in the section “Qualitative study” Participants for the focus group discussion were selected from existing socio demographic groups present in each district.

Comment 7: . Qualitative data analysis. Collection of qualitative data is described, but analysis is not well described beyond transcription, and key findings summarised. Was thematic coding or analysis conducted? What is meat by ‘analysed at different levels’?

Answer 7: We did not do thematic coding or analysis because we felt we might over-interpret the data recorded from the field. We plan to organize future study with a better design taking in consideration these structuring to further explore this aspect. 

Comment 8: Demographic differences. Although some demographic data is given about participants in the survey, the analysis does not differentiate between different cohorts of people. Were there differences between women and men, those with higher levels of education and lower, or by house type? If such analyses were not performed, or outcomes not deemed significant enough to report, the authors should state as such. Or perhaps, indicate that future publications might explore these differences. Indeed, generic conclusions can mask over important differences in knowledge, practices (and capacity) and perceptions that may have significant implications for who is most affected by malaria transmission.

Answer 8: These analysis were performed for some parameters but were not significant. A sentence addressing this point was added as a limit of the study in the discussion section. We have future studies in the pipeline that will address this point.

Comment 9: Quick clarifications. Line 86 – Sentence is not clear. What is meant by ‘the cost person’ as being ‘protected’? Perhaps the authors mean to say: The intervention is considered to be cost-effective, with costs per person being similar to those of LLIN use in urban settings’.

Answer 9: Corrected in the discussion section.

Comment 10: Long/confusing sentence. Line 107 – The sentence is long and confusing. I suggest rewording slightly (see below), and then removing the latter part of the sentence as the point about comparison between the intervention area and the non-intervention area is made in the next sentence. Note that ‘the larviciding trial’ should be replaced with ‘a larviciding trial’ as the specific trial being investigated in this paper has not yet been introduced or described. Suggested wording: In the present study, we tested the hypothesis that the implementation of a larviciding trial in the city of Yaoundé will improve knowledge and practices relating to malaria and mosquitoes, as well as increase people’s positive perceptions towards larviciding.

Answer 10: Corrected see the introduction.

Comment 11: Quick clarification. Line 149 – the authors describe the focus of the qualitative survey. It is unclear how 1) people’s knowledge and attitudes on preventive measures is different from 3) prevention measures. Do the authors mean the preventative practices reported as being undertaken by interviewees?

Answer 11: Corrected see line 166.

---

## [Editor Report · Decision Letter 2]

10 Oct 2022

Knowledge, practices and perceptions of communities during a malaria larviciding randomized trial in the city of Yaoundé, Cameroon

PONE-D-21-27631R2

Dear Dr. Antonio-Nkondjio,

We’re pleased to inform you that your manuscript has been judged scientifically suitable for publication and will be formally accepted for publication once it meets all outstanding technical requirements.

Kind regards,

Guangyu Tong

Academic Editor

PLOS ONE

Additional Editor Comments (optional):

We are pleased to accept your paper for publication.
---

## [Editor Report · Acceptance letter]

25 Oct 2022

PONE-D-21-27631R2 

Knowledge, practices and perceptions of communities during a malaria larviciding randomized trial in the city of Yaoundé, Cameroon 

Dear Dr. Antonio-Nkondjio:

I'm pleased to inform you that your manuscript has been deemed suitable for publication in PLOS ONE. Congratulations! Your manuscript is now with our production department. 

Kind regards, 

on behalf of

Dr. Guangyu Tong 

Academic Editor

PLOS ONE